# OpenReview forum: "$Q\sharp$: Provably Optimal Distributional RL for LLM Post-Training"
_ICML.cc/2025/Conference — Submitted to ICML 2025_

### Official Review · Reviewer_gBqj · 2025-03-10

**Overall Recommendation:** 4

**Summary:**

This work proposes Q♯, a distributional value-based algorithm for KL-regularized RL aimed at improving the post-training alignment of LLMs. The key idea is to learn the optimal Q function via a supervised distributional learning approach. The authors provide theoretical guarantees and validate their approach on both star-graph and real-world benchmarks (GSM8K and MATH). Experimentally, Q♯ demonstrates improved accuracy and a lower KL divergence relative to the reference policy.

**Claims And Evidence:**

The paper’s claims are supported by both theoretical analysis and empirical evidence.

**Essential References Not Discussed:**

While the paper cites many key works, it could benefit from referencing some of the most recent advances in stable distributional RL. Including such references would enrich the literature review and provide clearer context for the paper’s contributions.

**Experimental Designs Or Analyses:**

The experimental design is well-designed and includes both star-graph and real-world benchmarks (GSM8K and MATH). The analyses demonstrate the advantages of Q♯ in terms of accuracy and KL divergence. The suggestion is that the model used is the 8B model from the Llama 3 series. Could the authors conduct experiments with more models to enhance the generality of the approach? For example, using Llama 8B as the Q function model to guide Llama 70B, or using qwen 1.5B/3B to guide qwen 32B.

**Methods And Evaluation Criteria:**

The proposed method and evaluation criteria are suited for the problem at hand. Q♯ is designed  for LLM post-training under KL regularization, and the use of benchmarks such as GSM8K and MATH is appropriate for assessing math reasoning capabilities.

**Other Comments Or Suggestions:**

Minor improvements could be made in the clarity of the theoretical sections—especially for a broader audience not versed in RL theory. In addition, as mentioned in Weaknesses and Experiment sections, it is recommended that the authors conduct more extensive experiments to validate the universality of their approach.

**Other Strengths And Weaknesses:**

Strengths:

- Strong theoretical analysis with clear proofs and variance-dependent bounds.
- An innovative approach that avoids instability in TD learning is using supervised distributional methods.

Weakness

- The paper employs zero-shot prompts for evaluation; however, zero-shot prompting does not perform well on GSM8K and MATH for Llama 3 8B and Llama 3.1 8B, so it is recommended that the authors evaluate different prompting methods (like COT or TOT).

- The reliance on the deterministic MDP assumption may limit the applicability of the approach in settings with inherent stochasticity.

- Some parts of the theoretical exposition require additional explanation and more intuitive elaboration for readers less familiar with RL concepts.

**Questions For Authors:**

1. Can the authors elaborate on how Q♯ might be adapted to handle environments with stochastic transitions? Would the current theoretical guarantees extend, or are modifications necessary?


2. Could the authors provide more detailed comparisons of computational complexity and runtime performance between Q♯ and baseline methods?

**Relation To Broader Scientific Literature:**

The paper is well-contextualized within the broader RL and LLM post-training literature. It builds on foundational concepts from KL-regularized RL and distributional RL, and it addresses shortcomings in prior works (e.g., PPO, DPO, CD) regarding the correction of pre-training shortcuts.

**Theoretical Claims:**

I reviewed the proofs for Theorem 4.4. These proofs appear to be sound and are presented with sufficient rigor. Further clarification for readers less familiar with RL theory would be beneficial. The reliance on the deterministic MDP assumption should be discussed in terms of its limitations in more stochastic environments.

---

> ### Author Rebuttal · Authors · 2025-03-26
>
> Dear Reviewer gBqj,
>
> Thank you so much for your detailed review and suggestions for us to improve the paper. We respond to individual points below.
>
> **Could the authors conduct experiments with more models to enhance the generality of the approach?**
>
> **Authors' reply:** Thank you for the suggestion! Table 2 of the paper shows that both Llama 1B and Llama 3B can effectively guide Llama 70B, further improving its performance despite its strength over 8B models, supporting the generality of our approach.
>
> Additionally, as suggested by Reviewer GoFf, we tested Q# on a long-context benchmark, where it also performs well (see our response to Reviewer GoFf for details). We further extended experiments to the Qwen model series—please see the next point for results.
>
> **Evaluation with different prompting methods like COT or TOT.**
>
> **Authors' reply:** We conducted experiments using Qwen 2.5 1.5B to guide Qwen 2.5 7B model on GSM8K and MATH with COT prompting ("Let's think step by step"):
>
> |Dataset||GSM8K||
> |-|:-:|:-:|:-:|
> |Methods|$\pi^{\text{ref}}$|CD|$Q$#|
> |pass@1 ↑|76.1|79.0|**83.5**|
> |maj1@8 ↑|92.9|93.1|**93.8**|
> |KL-Divergence ↓|-|5.37|**4.10**|
>
> |Dataset||MATH||
> |-|:-:|:-:|:-:|
> |Methods|$\pi^{\text{ref}}$|CD|$Q$#|
> |pass@1 ↑|58.6|60.7|**61.9**|
> |maj1@8 ↑|72.8|74.2|**74.8**|
> |KL-Divergence ↓|-|7.07|**6.46**|
>
> This shows that Q# consistently outperforms baselines across both datasets while maintaining lower KL penalty, and highlights the generality of Q# across different model families and prompting methods. We will add these results in the final version.
>
> **Deterministic MDP assumption may limit applicability in settings with stochasticity. Can Q# be extended to MDPs with stochastic transitions?**
>
> **Authors' reply:** In this work, we focus on deterministic MDPs since the $Q^{\star,\eta}$ is a simple functional of the cumulative rewards distribution $Z^{\pi^{ref}}$, and that they capture the LLM post-training setting very well. If the MDP has stochastic transitions, then $Q^{\star,\eta}$ won't be a functional of $Z^{\pi^{ref}}$ (i.e. Thm 2.2 won't hold) and we will need to learn $Q^{\star,\eta}$ via temporal difference learning. Theoretically, this will require the bellman completeness assumption, which is much stronger than our realizability. Practically, TD-learning with function approximation is much less stable than online regression. Thus, while Q# can be extended to stochastic MDPs via TD-learning, this direction presents additional challenges and is less directly relevant to the LLM post-training application we focus on.
>
> **Could the authors provide more detailed comparisons of computational complexity and runtime performance between Q♯ and baseline methods?**
>
> **Authors' reply:** Q# and the CD baseline have the same computational complexity and runtime since both use the guidance model to compute a Q function alongside $\pi^{\text{ref}}$. Compared to generating responses solely with $\pi^{\text{ref}}$, Q# increases complexity by the ratio of the guidance model’s size to that of $\pi^{\text{ref}}$. For instance, guiding Llama 8B with Llama 1B increases complexity by ~12.5%.
>
> We efficiently implemented Q# in Hugging Face using LogitProcessor and key-value caches. On an Nvidia A6000, generating one response on test set of MATH takes 4.10s for $\pi^{\text{ref}}$ and 5.18s for Q#, slightly exceeding 12.5% possibly due to sequential Q function computation in LogitProcessor. We will open-source our implementation in the final version.
>
> **it could benefit from referencing some of the most recent advances in stable distributional RL.**
>
> **Authors' reply:** Thanks for the suggestion, we’ll include more references about DistRL in the final version. For example, [1] shows that the categorical DistRL loss, which we employ for our theory and experiments, enjoys smoothness and optimization stability under a bounded logit condition. [2] introduces a Sinkhorn DistRL loss which is a computationally efficient alternative for Wasserstein distance, and was shown to be more stable for multi-dimensional rewards. [3] proposed a KL-regularized categorical loss which they showed is empirically more stable in Atari games. However, these references all apply TD-learning with function approximation and replay buffers, which (Sutton and Barto, 2018) identified as a deadly triad that is notoriously difficult to scale, requiring many tricks such as double Q-learning and target networks. In contrast, our work obviates the need for TD-learning or tricks such as the target network by leveraging the special form of Q* in deterministic KL-regularized MDPs, which perfectly captures the LLM post-training application we focus on.
>
> [1] Sun et al. "How Does Return Distribution in Distributional Reinforcement Learning Help Optimization?."
>
> [2] Sun et al. "Distributional reinforcement learning with regularized wasserstein loss."
>
> [3] Li et al. "Efficient distributional reinforcement learning with Kullback-Leibler divergence regularization."

---

### Official Review · Reviewer_B1zk · 2025-03-13

**Overall Recommendation:** 4

**Summary:**

The paper introduces  a novel value-based reinforcement learning  algorithm designed for post-training LLM's. Unlike traditional policy-based methods like PPO and DPO, which may struggle with inherited pre-training biases, Q♯ leverages distributional RL to optimize a KL-regularized objective. It learns an optimal Q-function through distributional RL on an aggregated dataset, ensuring theoretical convergence to the optimal policy while maintaining low divergence from the reference model. The empirical results show a marked improvement over existing alignment techniques such as PPO and DPO. The theoretical analysis while novel seems quite limited. Moreover some terms crucial seem to be not well defined.

**Claims And Evidence:**

The paper claims that the  Q♯ method improves alignment performance over techniques such as DPO and PPO. It has provided benchmarks that showcase the significantly improved performance over the methods mentioned.

**Essential References Not Discussed:**

To my knowledge, I do not find any significant references that have been missed.

**Experimental Designs Or Analyses:**

I was not able to personally verify the code provided as I do not have access to the computational resources required.

**Methods And Evaluation Criteria:**

The usage of Llama 3 8b to showcase its improved performance over existing alignment methods is standard practice.

**Other Comments Or Suggestions:**

None

**Other Strengths And Weaknesses:**

The term $Reg_{mle}$ has not been formally defined. In context it is clear that it represents the regret but it should be formally defined.

**Questions For Authors:**

None

**Relation To Broader Scientific Literature:**

The paper provides an improved post training method as compared to existing ones such as PPO (Schulman et al., 2017) and DPO (Rafailov et al., 2023). The theoretical basis of this is based on providing a convergence proof for a regularized KL divergence setup where the deterministic nature of the setup is used to cast the problem as an online learning problem (Orbana, 2019).

**Theoretical Claims:**

I have checked the theoretical claims. I did not find any major errors.

I have only a minor mistake(I think) on page 13.
On the right hand side of the first equation should the second term not be
$Q^{\pi^{k},\eta}_{h} $

  instead of  $Q^{*,\eta}_{h}$

---

> ### Author Rebuttal · Authors · 2025-03-26
>
> Dear Reviewer B1zk,
>
> Thank you for your encouraging review and strong support! We address individual questions below.
>
> **On the right hand side of the first equation should be $Q^{\star,\eta}_h$.**
>
> **Authors' reply:** Thanks for catching this typo! We will make sure to fix it in the final version of the paper.
>
> **The term $Reg_{mle}$ has not been formally defined. In context it is clear that it represents the regret but it should be formally defined.**
>
> **Authors' reply:** The reviewer is fully correct that $Reg_{mle}$ represents the regret of the MLE oracle after K iterations, which is currently defined in Line 377. We will make sure to highlight this definition in the revision.
>
> Please let us know if you have any other questions!

---

### Official Review · Reviewer_GoFf · 2025-03-14

**Overall Recommendation:** 4

**Summary:**

The paper introduces a value-based algorithm for KL-regularized RL in deterministic MDP, where the authors leverage distributional RL to estimate the optimal regularized $Q$-function. Theoretically, the authors reduce KL-regularized RL to no-regret online learning, proving variance-dependent PAC bounds under realizability assumptions.

**Claims And Evidence:**

The main claims (convergence & optimality) are supported well by the theoretical and empirical results. However, the authors claim "the policy-based methods can suffer from certain shortcuts that are picked up during pre-training, which fail to generalize for tasks that involve planning." They simply use a special case - in star-graph tasks - to support this claim, which seems insufficient.

**Essential References Not Discussed:**

The key contribution is the approach for KL-regularized RL. The paper only cites parts of existing works for the same problem, but misses a number of literatures like [Nachum'19a][Nachum'19b].

Reference:

[Nachum'19a] AlgaeDICE: Policy Gradient from Arbitrary Experience.

[Nachum'19b] DualDICE: Behavior-agnostic Estimation of Discounted Stationary Distribution Corrections.

**Experimental Designs Or Analyses:**

The main experimental design seems sound, and the ablation results are good. Yet, it would be better if the authors could include the convergence curves for better persuasion.

**Methods And Evaluation Criteria:**

The use of distributional RL to estimate the cumulative reward distribution makes sense, which can avoid TD instability. The used benchmarks and metrics are standard and commonly used, but comparisons with policy-based methods like AlgaeDICE [Nachum'19] are missing.

Reference:

[Nachum'19] AlgaeDICE: Policy Gradient from Arbitrary Experience.

**Other Comments Or Suggestions:**

C1: Advantages over policy-based methods could be further explored and discussed.

C2: There are some missing literatures.

C3: The introduction could be further improved to be more straightforward.

**Other Strengths And Weaknesses:**

This paper offers a new value-based approach for KL-regularized RL, building on distributional RL. The main claims are supported by the theoretical and empirical results. The paper also provides some good insights into the introduced method.

However, the paper reducing the KL-regularized RL to deterministic RL weakens its technical contribution. Its motivation - the value-based methods excel the policy-based counterparts - seems not very clear. In addition, the work does not delve into more unique properties of LLM post-training (like long contexts), making its relation to LLM not very strong.

**Questions For Authors:**

Q: Can the proposed approach be extended to stochastic MDPs?

---

After rebuttal, the authors have addressed most of my concerns. While the underlying reason of Q#'s superiority over the policy-based counterparts seems still unclear in the cases beyond 'star-graph', I believe the proposed methodology is instructive for the community.

I have updated my score to 4.

**Relation To Broader Scientific Literature:**

The work builds on distributional RL to address the KL-regularized objective. It addresses the suboptimality of existing counterparts like CD and VAS.

**Theoretical Claims:**

I went through the proof of the Theorem 4.4 but didn't check all details. It exploits the Hellinger distance to build the relation between the optimal and learned $Q$-functions and Multiplicative Azuma's inequality to bound the sum of squared Hellinger distances. The result seems technically sound.

---

> ### Author Rebuttal · Authors · 2025-04-01
>
> Dear Reviewer GoFf,
>
> Thank you for the detailed review and constructive feedback! We truly appreciate the time and effort that have been invested in providing constructive comments. Please find our responses below.
>
> **Star-graph results for claiming policy-based method shortcomings in planning tasks**
>
> **Authors' reply:** We appreciate the reviewer’s feedback and agree that our claim about policy-based methods picking up shortcuts should be framed more narrowly. Our findings are specific to the star-graph task, where we observe that policy-based methods tend to exploit pretraining shortcuts rather than learning effective planning strategies. While this provides concrete evidence of the issue in this setting, we acknowledge that further investigation is needed to generalize this claim beyond the studied task. We will clarify this limitation in the final version of the paper.
>
> **Additional related work such as [Nachum'19a][Nachum'19b]**
>
> **Authors' reply:** Thank you for pointing out DICE, which we’ll certainly add a discussion and citation in our final version. DualDICE (for off-policy evaluation [Nachum'19a]) and AlgaeDICE (for off-policy learning [Nachum'19b]) are motivated by the curse of horizon in offline RL where the variance of importance weighting grows exponentially in $H$. DICE obviates this explosion in variance by Lagrangianizing the linear program of $Q^\pi$ and fitting an additional nuisance function $w$, which is the density ratio between the visitation measure under $\pi$ and the data distribution $\nu$. DICE was shown to be effective in low-dimensional control tasks [Nachum’19a, Nachum’19b], although learning the $w$ function was also shown to be difficult in high-dimensional control tasks [Chang’22]. Exploring DICE for LLM post-training is a promising direction to pursue for future work and is beyond the scope of our paper.
>
> **the work does not delve into more unique properties of LLM post-training (like long contexts)**
>
> **Authors' reply:** Thank you for the valuable comment! Following your suggestion, we conducted experiments on a long context benchmark, QuALITY [Pang’22], a multiple-choice QA dataset with context passages in English sourced from Project Gutenberg. We experimented with two settings: Qwen 2.5 1B guiding Qwen 2.5 7B and Llama 3.2 1B guiding Llama 3.1 8B.
>
> |$\pi^{\text{ref}}$||Qwen 2.5 7B|||Llama 3.1 8B||
> |-|:-:|:-:|:-:|:-:|:-:|:-:|
> |Methods|$\pi^{\text{ref}}$|CD|$Q$#|$\pi^{\text{ref}}$|CD|$Q$#|
> |pass@1 ↑|64.5|64.2|**68.1**|73.5|75.1|**75.9**|
> |maj1@8 ↑|72.0|66.3|**73.3**|79.3|79.3|**81.1**|
> |KL-Divergence ↓|-|12.32|**7.90**|-|9.23|**8.88**|
>
> The results suggest that Q# can also consistently improve performance on long context tasks for models such as Qwen 2.5 and Llama 3.1 with smaller KL divergence than CD baseline.
>
> Additionally, we conducted new experiments on the star-graph $G(2, 20)$ to stress-test long context planning capabilities, where the LLM has to perform 20 backup reasoning steps to solve the task. We observe that Q# solves $G(2, 20)$ with nearly 100% accuracy while all policy-based post-training approaches get stuck at 50% accuracy and fail to fix the short-cut, which matches our existing results on $G(2,5),G(5,5)$ and $G(3,8)$ in the paper. We will add these new results to the final version of the paper.
>
> **Can the proposed approach be extended to stochastic MDPs?**
>
> **Authors' reply:** In this work, we focus on deterministic MDPs since the $Q^{\star,\eta}$ is a simple functional of the cumulative rewards distribution $Z^{\pi^{ref}}$, and that they capture the LLM post-training setting very well. If the MDP has stochastic transitions, then $Q^{\star,\eta}$ won't be a functional of $Z^{\pi^{ref}}$ (i.e. Thm 2.2 won't hold) and we will need to learn $Q^{\star,\eta}$ via temporal difference learning. Theoretically, this will require the bellman completeness assumption, which is much stronger than our realizability. Practically, TD-learning with function approximation is much less stable than online regression. Thus, while Q# can be extended to stochastic MDPs via TD-learning, this direction presents additional challenges and is less directly relevant to the LLM post-training application we focus on.
>
> Additional Citations:
>
> [Nachum'19a] AlgaeDICE: Policy Gradient from Arbitrary Experience.
>
> [Nachum'19b] DualDICE: Behavior-agnostic Estimation of Discounted Stationary Distribution Corrections.
>
> [Chang’22] Learning bellman complete representations for offline policy evaluation.
>
> [Pang’22] QuALITY: Question Answering with Long Input Texts, Yes!

---

> > ### Comment · Reviewer_GoFf · 2025-04-08
> >
> > Thanks for the rebuttal. After re-evaluating the manuscript and reading other comments, I still have the following concerns/questions:
> >
> > 1. I am not convinced why the proposed method is superior to the existing policy-based method in the domains beyond the "star-graph" cases.
> >
> > 2. After finding the optimal parameters of $Z$, how to instantiate the learning policy in practice?
> >
> > 3. Can the authors shed more light on the meaning of "shortcuts"? Why the proposed method can alleviate it?
> >
> > 4. Can you add more details on the derivation from Eq. (3) to Eq. (4)? Should not $V^{*,\eta}_{h}$ correspond to the optimal policy?

---

> > > ### Author Response · Authors · 2025-04-09
> > >
> > > Dear Reviewer GoFf,
> > >
> > > Thank you for your valuable feedback and questions. We truly appreciate the opportunity to answer your remaining questions:
> > >
> > > **Q1.**
> > >
> > > Thanks for the question. We only intend to claim that Q♯ outperforms policy-based methods on the star-graph task. We apologize for the confusion and will revise the text to clarify this positioning.
> > >
> > > We chose star-graph to highlight a known failure case of next-token prediction loss [1]. As shown in [2], it embeds the "sparse parity" problem — determining whether the sum of a binary string is even or odd — which is known to be difficult for gradient-based optimizers and is widely studied in learning theory and optimization [3,4,5,6]. Though the task is simple, it captures an important failure case for policy-based methods that struggle to generalize to unseen graphs due to learning shortcuts (please see our response to Q3 for discussion on shortcuts). In contrast, we show that value-based methods like Q♯ do not learn shortcuts and generalizes well at test time.
> > >
> > > Beyond star-graph, a practical benefit of Q♯ is that it allows training a small guiding model while keeping the reference policy frozen. As shown in Table 2, a 1B Q♯ model can effectively steer a 70B reference model. Post-training this 70B model is far more expensive using policy-based methods like REINFORCE or DPO and is infeasible with the compute resources available in our academic setting.
> > >
> > > **Q2.**
> > >
> > > After estimating the parameters of $Z$ — the conditional distribution of cumulative rewards under $\pi_{ref}$ — we instantiate the policy via $\pi(y\mid x) \propto \pi\_{ref}(y\mid x) \cdot \mathbb{E}_{z\sim Z(x,y)}[\exp(z/\eta)]$, as described in Eq. (5) and implemented in Line 4 of Algorithm 1. This is motivated by the fact that, if we recover the true distribution $Z^\star$, this recovers the optimal KL-regularized policy $\pi^{\star,\eta}$ defined in Eq. (2). Moreover, Theorem 4.4 provides a bound on the regret incurred from using an approximate $Z$ learned from data, showing that the induced policy converges to $\pi^{\star,\eta}$. We’ll include an explicit definition of the induced policy within the algorithm box to make this clearer in the revision.
> > >
> > > **Q3.**
> > >
> > > By “shortcuts,” we refer to spurious behaviors that arise during pre-training with the auto-regressive next-token prediction. This is also called the Clever Hans Trick by [1], because the model simply learns to follow the next edge as opposed to solving the actual star-graph task. The consequence of this shortcut is that models achieve low training loss by memorizing the first token in the training set and following the Clever Hans Trick, but struggle at test time when the first token is not available. Thus, the model trained with next-token prediction loss fails to generalize on test graphs and achieves poor accuracy. Q♯ avoids this failure mode by **not relying on next-token supervision**. It learns to predict the future cumulative rewards (i.e., reward-to-go) under the reference policy and uses that to guide generation.
> > >
> > > **Q4.**
> > >
> > > Thank you for the helpful question! We’re happy to clarify this derivation.
> > >
> > > Eq. (3) expresses the soft value function under the optimal policy as: $\exp(\eta^{-1}V^{\star,\eta}\_h(x))=\mathbb{E}\_{y\sim\pi\_{ref}(x)}[\exp(\eta^{-1}r\_h(x,y)+\eta^{-1}V^{\star,\eta}\_{h+1}(x’))]$, where $x'=P(x, y)$ is the next state. For LLMs, $x’$ is the concatenation of $x$ and $y$. This follows from the log-sum-exp form of soft value iteration (Eq. (2)), where the optimal policy is an exponential tilting of $\pi\_{ref}$.
> > >
> > > We recursively apply this to $V^{\star,\eta}\_{h+1}(x’)$ and so on, unrolling until the final step. Each stage adds a new reward term inside the exponent, and since exponentials multiply, they combine into an exponent of the reward sum — yielding Eq. (4).
> > >
> > > Although $V^{\star,\eta}\_{h}(x’)$ corresponds to the soft value of the optimal policy, its *recursion* is expressed via expectations over $\pi\_{ref}$, since the optimal policy is a softmax over $\pi\_{ref}$ weighted by exponentiated cumulative rewards (Eq. (2)). We’ll include this clarification in the revision.
> > >
> > > Thanks again for your valuable comments. We hope we have addressed your concerns and we will certainly include all of the above discussion and new experimental results into the final version!
> > >
> > > **Citations**
> > >
> > > [1] Bachmann and Nagarajan, “The pitfalls of next-token prediction”, ICML 2024.
> > >
> > > [2] Hu et al, “The Belief State Transformer”, ICLR 2025.
> > >
> > > [3] Shalev-Shwartz et al, “Failures of Gradient-Based Deep Learning”, ICML 2017.
> > >
> > > [4] Barak et al, “Hidden Progress in Deep Learning: SGD Learns Parities Near the Computational Limit”, NeurIPS 2022.
> > >
> > > [5] Kou et al, “Matching the Statistical Query Lower Bound for k-Sparse Parity Problems with Sign Stochastic Gradient Descent”, NeurIPS 2024.
> > >
> > > [6] Abbe and Sandon, “Polynomial-time universality and limitations of deep learning”, Communications on Pure and Applied Mathematics 2023.

---

### Decision · Program_Chairs · 2025-05-01

**Decision:**

Reject

**Comment:**

This work proposes a value-based distributional RL algorithm and applies it to LLM fine-tuning. The key innovation is to incorporate KL divergence-based regularization into the likelihood objective, and use this to enforce proximity to a prior distribution, if available. One of the key innovations is introducing distributional RL into the fine-tuning setting, and demonstrating experimentally it outperforms several standard benchmarks, including DPO.

During the discussion phase, the reviewers brought out the importance of comparison to methods that incorporate divergence to a prior:

[Nachum'19a] AlgaeDICE: Policy Gradient from Arbitrary Experience.

[Nachum'19b] DualDICE: Behavior-agnostic Estimation of Discounted Stationary Distribution Corrections.

which the authors have added in the rebuttal phase. However, from current batch of reviews and responses, there is no discussion of the novelty of Theorem 4.4 -- whether only assumptions on realizability are actually more relevant for the application domain under consideration, or whether the specific form of the second-order bound has usable insights for practical algorithm design in any of the experiments or explains the performance improvements the authors observe in practice. Most importantly, the algorithmic formulation posed here mostly pre-dates this work and the LLM application. For these reasons, the paper is not acceptable in its current form, as the innovations that distinguish it from prior art in RL are insufficient.